# Acute *Amanita muscaria* Toxicity: A Literature Review and Two Case Reports in Elderly Spouses Following Home Preparation

**DOI:** 10.3390/toxins17120570

**Published:** 2025-11-25

**Authors:** Stanila Stoeva-Grigorova, Ivanesa Yarabanova, Maya Radeva-Ilieva, Diana Ivanova, Snezha Zlateva, Petko Marinov

**Affiliations:** 1Department of Pharmacology, Toxicology and Pharmacotherapy, Faculty of Pharmacy, Medical University of Varna, 9000 Varna, Bulgaria; maya.radeva@mu-varna.bg (M.R.-I.); snezha.zlateva@mu-varna.bg (S.Z.); petko.marinov@mu-varna.bg (P.M.); 2Clinical Toxicology Department, Naval Hospital, 9000 Varna, Bulgaria; ivanesa_98@abv.bg; 3Department of Biochemistry, Molecular Medicine and Nutrigenomics, Faculty of Pharmacy, Medical University of Varna, 9000 Varna, Bulgaria; divanova@mu-varna.bg

**Keywords:** *Amanita muscaria*, fly agaric, intoxication, poisoning, mushrooms, ibotenic acid, muscimol

## Abstract

*Amanita muscaria (L.) Lam*., commonly known as fly agaric, remains an uncommon yet clinically important cause of acute mushroom intoxication. Although typically associated with mild to moderate neuropsychiatric disturbances, the mushroom’s toxic profile is highly variable and continues to attract scientific, toxicological, and public health interest. This work provides an integrative review of the biochemical composition, toxicodynamics, and clinical manifestations associated with *A. muscaria* exposure, with particular emphasis on the pharmacological actions of its principal constituents, ibotenic acid and muscimol. The review is complemented by two contemporaneous cases of severe intoxication in elderly individuals, illustrating the real-world clinical expression of the toxidrome and the challenges in diagnosis and management. Both cases presented with rapid-onset gastrointestinal symptoms, profound central nervous system depression, and cholinergic features, requiring intensive supportive therapy, atropine infusion, and continuous monitoring. Full recovery was achieved in both patients. These clinical observations contextualize the broader toxicological framework discussed in the review and underscore the need for increased clinical vigilance, improved public education, and strengthened diagnostic and therapeutic preparedness regarding psychoactive wild mushroom exposures.

## 1. Introduction

The annual global mortality rate due to mushroom poisoning remains unknown. There are notable regional differences in the incidence of these intoxications worldwide, but high-quality epidemiological data are scarce [1]. In Europe alone, approximately 50–100 fatal cases are reported each year. In this context, the emerging risk of mushroom poisoning linked to migratory processes has been highlighted. Apparently, some migrants collect unfamiliar wild mushroom species for food due to adverse economic conditions, thereby increasing the likelihood of intoxications [2]. In the United States, the consumption of wild and potentially toxic mushrooms is also a common practice. Between 1999 and 2016, 133,700 poisoning cases were recorded, although severe intoxications were rare—only 704 cases—with 52 fatalities, predominantly resulting from cyclopeptide-containing mushrooms in adults. The primary cause of these incidents is the misidentification of edible species, a problem that, according to Brandenburg and Ward (2018), can be mitigated through proper education and public awareness [3]. According to Goldfrank (2019) [4], unintentional mushroom ingestion, particularly in children, represents a minor fraction of all exposures reported to poison control centers—less than 0.25% of total cases. Most mushroom intoxications are asymptomatic or minimally toxic, primarily affecting adults, and are associated with Amanita species, as well as certain hallucinogenic mushrooms or those containing gyromitrin. Alarmingly, the exact species ingested remains unidentified in 75–95% of cases [4].

*Amanita muscaria* (*L.*) *Lam.* (*A. muscaria*) belongs to the phylum Basidiomycota, order Agaricales, and family Amanitaceae. The species exhibits considerable phenotypic diversity, particularly in cap coloration; nonetheless, it is most readily recognized by its bright red cap adorned with white warts—a visual hallmark that has become iconic in both mycology and religious symbolism (Figure 1). For centuries, the mushroom has been known in various cultures as fly agaric or fly amanita due to its traditional use in milk to attract and sedate flies through its bioactive compounds [5].

Ethnomycological evidence suggests that *A. muscaria* played a central role in ancient entheogenic traditions. It was reportedly used in the mystery cults of Dionysus in Greece and Mithras in Rome to elicit ecstatic and transcendental states. In Siberia, the mushroom held sacred status and was integral to shamanic rites long before alcohol became widespread [6]. Its distinctive pharmacokinetics—especially the urinary excretion of largely unmetabolized ibotenic acid and muscimol—enabled the practice of “secondary intoxication”, in which shamans or followers consumed the urine of humans or reindeer that had ingested the mushroom to enhance psychoactive effects while reducing toxicity. Some hypotheses further propose that *A. muscaria* may correspond to the mythical Soma of the *Rigveda*, the divine elixir believed to grant immortality and healing powers [6,7,8].

Despite its known raw toxicity, *A. muscaria* remains a food source in some regions, reflecting both its biochemical complexity and cultural persistence. Traditional preparations in Mexico and Italy rely on boiling and discarding the toxin-rich water, while in North America, the mushroom is dried and smoked after removing the red cuticle [9]. In Japan, a more prolonged process is applied, involving drying, brining for approximately twelve weeks, and repeated rinsing prior to consumption [8]. The mushroom’s symbolic and mythological resonance endures. It continues to appear in ritualistic and spiritual practices in the United States and Europe, amid growing public interest driven by changing attitudes toward psychoactive substances and widespread online information [10,11,12,13]. Although *A. muscaria* occupies a regulatory “gray zone”, Savickaitė and Laubner-Sakalauskienė (2025) classify it, under certain conditions, as a novel psychoactive substance (NPS) [14]. While NPSs are mostly synthetic, the category also includes natural compounds outside international control frameworks. The European Food Safety Authority (2023) has raised concerns about increasing use of herbal alcohol substitutes acting on the GABA system and highlights rising *A. muscaria* consumption as an emerging public health risk [14]. Consequently, several European countries (Romania, the Netherlands, Poland, Lithuania) have classified it as a controlled substance and/or prohibited its sale [14,15]. Within this context, scientific debate persists regarding whether *A. muscaria*’s reputation as a “deadly poison” is overstated and whether restrictions outside research settings are warranted [10,13,16].

Given rising public interest in wild mushroom foraging and renewed entheogenic practices, *A. muscaria* intoxications are becoming increasingly relevant to clinical and public health practice. This trend highlights the need to update medical knowledge, improve clinical vigilance, and strengthen toxicological preparedness for recognizing and managing these uncommon but potentially serious poisonings [4,17]. This study presents current data on *A. muscaria* toxicity and reports two contemporaneous cases within the same household—one male and one female—both resulting in complete recovery. These cases serve two purposes: they broaden the understanding of the clinical presentation and progression of *A. muscaria* poisoning, and they provide a basis for discussing contemporary diagnostic and therapeutic strategies. The work also synthesizes current knowledge on the pathophysiology and pharmacokinetics of the principal toxins, ibotenic acid and muscimol, and outlines the spectrum of clinical effects, from gastrointestinal symptoms to neurological and psychotropic manifestations.

## 2. Biochemical Composition and Bioactive Constituents of *A. muscaria*

*A. muscaria* contains a complex ensemble of bioactive compounds capable of producing diverse neurological and somatic effects, influenced by dose, route of exposure, and individual susceptibility. Among these constituents, the alkaloid muscarine is notable for its historical misattribution as the mushroom’s principal toxin (Figure 2A). Later research, however, established that *Inocybe* species contain far higher muscarine levels and generate toxic effects that differ fundamentally from those of *A. muscaria*. Crucially, *Inocybe* poisonings do not produce hallucinations or other central neurological symptoms typical of fly agaric, thereby excluding muscarine as its primary psychoactive agent [6].

Chemically, muscarine is a quaternary ammonium alkaloid first isolated in pure form by King in 1922, with its structure elucidated by Keogh et al. in 1957 [18]. It is present only in trace amounts in *A. muscaria*, primarily within the basidiocarps (~0.02% of dry weight), accounting for its minimal contribution to overall toxicity. Despite its low abundance, muscarine represents a compelling toxicological model due to its pharmacodynamic properties [19]. It is water-soluble and thermally stable, retaining activity upon culinary processing [1]. Pharmacologically, muscarine functions as a non-selective agonist of muscarinic acetylcholine receptors within the parasympathetic nervous system. Its partial resistance to cholinesterase-mediated degradation prolongs its action. The presence of a positively charged quaternary amine prevents muscarine from crossing the blood–brain barrier, thereby precluding direct central nervous system effects [18]. Clinically, muscarine predominantly mediates peripheral autonomic symptoms in *A. muscaria* poisoning, including hypersalivation, hyperhidrosis, bradycardia, and diarrhea, typically mild and reversible, but capable of inducing severe cholinergic crises at high doses. Consequently, muscarine plays an ancillary role in the overall toxicological profile, while central effects are chiefly mediated by ibotenic acid and muscimol (Figure 2B,C) [10]. The red cuticle and underlying yellowish tissue of the mushroom contain the highest concentrations of these compounds [20]. Approximately 10–20% of ingested ibotenic acid is metabolized to muscimol in the gastrointestinal tract. Ibotenic acid crosses the blood–brain barrier to reach the CNS, while the remainder circulates systemically and is excreted in urine within hours [21]. It exerts excitatory effects primarily via NMDA-glutamate receptor activation, intimately linked to memory, learning, and synaptic plasticity. Intracerebral administration in animal models induces clinical and histopathological changes analogous to Alzheimer’s disease, rendering it a valuable model for neurodegenerative research [22,23]. Muscimol, in turn, is a tenfold more potent agonist of pre- and postsynaptic GABAA receptors and readily crosses the blood–brain barrier. Its action opens associated chloride channels, hyperpolarizing neuronal membranes and suppressing neuronal activity. Clinically, muscimol induces sedation, visual distortions, hallucinations, ataxia, postural instability, muscle fasciculations (indicative of neuromuscular involvement and early motor dysregulation potentially progressing to generalized seizures), as well as nausea and vomiting. Rapid metabolism of muscimol post-ingestion implies that toxic effects result from both the parent compound and its psychoactive metabolites; however, detailed metabolic studies on muscimol or ibotenic acid remain unpublished [18].

Active compound content is variable, but a single *A. muscaria* fruiting body (50–70 g fresh weight) typically contains approximately 6 mg muscimol and up to 70 mg ibotenic acid—sufficient to elicit pronounced psychoactive effects in an adult [9,12]. In addition to these two primary isoxazole derivatives, muscazone, a photochemically derived product of ibotenic acid under ultraviolet irradiation, has been identified, though it exhibits comparatively weak pharmacological activity [9]. Other bioactive constituents include muscaridine, stizolobic and stizolobinic acids, various pigments, and trace amounts of tropane alkaloids. Furthermore, *A. muscaria* exhibits bioaccumulation properties, concentrating vanadium and other toxic metals from soil, potentially modifying its toxicological profile depending on environmental growth conditions [6,8]. According to Ordak et al. (2023), additional toxic elements, such as mercury, may contribute to the manifestation of visual and auditory hallucinations, particularly given that *A. muscaria* grows in the upper soil layers [16].

## 3. Acute Poisoning and Toxicodynamics of *A. muscaria*

Fatal intoxications with ibotenic acid and muscimol are exceedingly rare, as both mycotoxins exhibit lethal effects only at substantially high doses. The established oral LD_50_ in rats is approximately 129 mg/kg for ibotenic acid and 45 mg/kg for muscimol, corresponding to their relatively low acute toxicity in humans [18,24]. Another *Amanita* species, *A. pantherina* (family Amanitaceae), contains similarly high concentrations of ibotenic acid and muscimol. Consequently, the clinical syndrome induced by the ingestion of mushrooms containing these two mycotoxins is termed “pantherina syndrome” (or pantherina-muscaria syndrome) (Figure 3) [18,20,25]. It is also referred to as “mycoatropine” in the literature, as the symptoms resemble those caused by atropine-containing plants such as *Datura stramonium*, *Atropa belladonna*, and *Hyoscyamus niger*, although tropane alkaloids are absent [7].

The syndrome is characterized by a short latent period, typically 30 min to 2–3 h post-ingestion [17,26]. Initial manifestations include nausea, vomiting, and diarrhea, rapidly followed by neurological symptoms reflecting central nervous system dysfunction, such as confusion, vertigo, myoclonus, visual and auditory hyperesthesia, and distorted perception of time and space, accompanied by mydriasis, lethargy, and deep sleep [7,27,28]. Hallucinations usually emerge within two hours, persist for up to eight hours, and progress through an excitatory phase (warmth, tingling, restlessness, hyperactivity, ataxia, floating sensations, visual and auditory hallucinations) followed by deep sleep, from which patients often awaken spontaneously with neuromuscular hyperactivity and hypotension. Total intoxication duration is roughly 24 h, and many individuals describe a subjective sense of “rebirth” upon recovery [6,18]. Severe cases may present with tremors or tonic–clonic seizures, loss of consciousness, marked mydriasis, absent superficial and deep reflexes, and life-threatening respiratory or cardiovascular compromise. Muscarinic-like symptoms, such as excessive salivation and sweating, may also occur but are generally transient. Retrograde amnesia is common, and symptom severity correlates with ingested mushroom quantity [6,20]. This mycotoxicosis syndrome is frequently misdiagnosed as alcohol intoxication due to early behavioral similarities [18].

Mortality from accidental *A. muscaria* poisoning is exceedingly low, ranging between 2% and 5% of reported cases [6]. Clinical manifestations typically resolve within a few hours. Given that the intoxication profile of *A. muscaria* combines both cholinergic and anticholinergic effects, no specific antidote exists. Management is primarily symptomatic and includes:∘Supportive therapy to maintain respiratory and cardiovascular function;∘Control of agitation and seizures using benzodiazepines;∘Gastrointestinal decontamination with activated charcoal when presented early;∘Close monitoring in an intensive care setting, as the patient’s condition may fluctuate between excitatory and depressive phases, or hospitalization with neurological observation;∘Correction of fluid and electrolyte imbalances in cases of severe dehydration (Table 1) [4,6,18,27,28,29].

The toxic effects of muscarine, present in minimal amounts in *A. muscaria*, are dose-dependent. Its established intravenous LD_50_ in mice is 0.23 mg/kg. Although muscarine exerts parasympathomimetic effects—including salivation, sweating, bronchospasm, bradycardia, and hypotension—intoxications are rarely fatal. Children and elderly patients, particularly those with preexisting cardiovascular conditions, are more susceptible [6]. Unlike many mycotoxins, muscarine has a specific antidote, atropine. However, in clinical practice, atropine use in *A. muscaria* intoxications is limited and must be carefully considered, as it antagonizes only peripheral muscarinic effects and does not counteract the central manifestations induced by ibotenic acid and muscimol, which act via NMDA- and GABAA-mediated mechanisms. Moreover, inappropriate administration of atropine may complicate the clinical course or mask critical symptoms; therefore, its use should be individualized based on clear clinical indication and under appropriate monitoring [18,30,31].

## 4. Clinical Cases

We report two analogous and contemporaneous cases of acute mushroom poisoning in elderly spouses—a 79-year-old male and a 78-year-old female. Both patients had no significant history of chronic diseases or comorbid conditions that could have materially influenced the course of intoxication. The poisoning occurred following the ingestion of wild *A. muscaria* mushrooms, collected and prepared at home.

The initial clinical manifestations appeared approximately 30 min after ingestion, representing an exceptionally short latent period indicative of rapid absorption and potent toxic effects of the consumed mushrooms. Recognizing the link between their symptoms and mushroom consumption, both patients promptly sought medical attention at the Emergency Medical Center in Byala (Varna region, Bulgaria). Deterioration of their condition necessitated transport and hospitalization to the Clinical Toxicology Department at Naval Hospital—Varna, Bulgaria Medical Academy (50–60 km away). Significant clinical worsening occurred during transport.

Initially, both patients experienced pronounced somnolence, nausea, and general malaise, followed by vomiting that intensified, becoming profuse and repetitive. Within a short interval, rapid progression to impaired consciousness occurred, advancing to sopor and subsequent coma. On admission, the clinical picture was severe: impaired consciousness (Glasgow Coma Scale score 6), pupils of normal diameter with sluggish light reaction, foamy oral secretions, moderate salivation, continuous vomiting, tachypnea (20/min) with upper airway rales, and perioral cyanosis despite relatively stable hemodynamics. Fibrillar limb fasciculations were observed; the female exhibited urinary incontinence, while the male demonstrated quadriparesis, quadrireflexia, and absent abdominal reflexes. Hospitalization was initiated approximately 2 h and 30 min after mushroom ingestion.

### 4.1. Objective Findings

Upon admission, both patients were in a severely compromised state, presenting with profound coma and the characteristic neurological and somatic disturbances associated with mushroom intoxication. In addition to the deep comatose state with persistent depression of consciousness, the following signs were observed in the subsequent hours: pronounced hypersalivation, repetitive fibrillar limb fasciculations, and gradually developing miosis with sluggish pupillary reactivity. In the female patient, a generalized tonic–clonic seizure occurred approximately one hour after the administration of atropine.

### 4.2. Laboratory Findings

At admission, the male patient’s liver enzyme levels (AST, ALT, GGT) were within the reference ranges and even demonstrated a slight decrease at discharge, with no evidence of a cytolytic syndrome. His total bilirubin was elevated at admission (31 µmol/L; reference upper limit 21 µmol/L) but declined to 24 µmol/L by discharge, indicating transient hyperbilirubinemia likely of functional origin, associated with the toxic effects of fungal metabolites and hemodynamic changes rather than permanent hepatic injury. Cholinesterase, while within the variable reference range, showed a decreasing trend—from 8.8 IU/mL at admission to 7.1 IU/mL—typical for intoxications with cholinergic manifestations, reflecting either hepatic synthetic impairment or direct enzymatic inhibition. Renal function parameters, including urea, creatinine, and eGFR, were normal at admission.

In the female patient, a mild increase in AST (23 → 34 U/L) and minimal elevation in ALT (11 → 14 U/L) was observed without clinical significance. GGT remained low and stable, and bilirubin stayed within normal limits, slightly decreasing over time, excluding significant hepatocellular injury. Notably, cholinesterase declined markedly from 7.5 to 5.3 IU/mL, correlating with more severe clinical manifestations, including seizures and hallucinations, indicating stronger systemic toxic effects. Renal monitoring at admission revealed elevated urea and serum creatinine, with eGFR measured at 36 mL/min, confirming impaired renal function. During therapeutic intervention, clear improvement was observed, consistent with transient and reversible renal impairment, likely secondary to the toxin-induced dysmetabolic effect and severe gastrointestinal losses (vomiting and dehydration). Table 2 summarizes the laboratory parameters for both patients at admission and discharge.

### 4.3. Imaging Findings

The imaging studies performed in the male patient revealed congested pulmonary hila, most likely reflecting transient circulatory stasis within the pulmonary microcirculation (Figure 4A). Such a finding may result from temporary cardiovascular compromise, including hemodynamic alterations induced by severe intoxication, aggressive intravenous fluid therapy, and forced diuresis [32,33,34,35]. The absence of definitive infiltrative lesions argues against a pneumonic process, suggesting functional-dynamic changes rather than structural damage [36]. In the female patient, imaging indicated minimal pleural effusions without evidence of pulmonary infiltrates (Figure 4B). This observation can be interpreted as a reactive pleural response, likely related to hydrostatic imbalance or increased capillary permeability secondary to systemic intoxication. The absence of infiltrates supports the conclusion that these findings do not represent bacterial infection but rather a secondary, transient manifestation of severe toxic and metabolic stress [37].

### 4.4. Differential Diagnosis and Applied Diagnostic Criteria

These two cases of acute *A. muscaria* poisoning—characterized by rapid onset of coma, hypersalivation, miosis, fasciculations, seizures, and progressive neurological decline within hours—are consistent with a cholinergic-toxic syndrome from muscarine and related compounds. The exceptionally short latent period (~30 min) indicates rapid absorption and potent toxicity. Key differential considerations are:∘Other toxic mushrooms—A. phalloides (delayed gastrointestinal symptoms, hepatotoxicity, rare early CNS depression); *Inocybe* spp., and *Clitocybe* spp. (cholinergic signs without prominent CNS effects or hallucinations);∘Non-mushroom cholinergic toxins—Organophosphate or carbamate exposure (requires history and additional systemic features);∘Neurological/metabolic disorders—Acute cerebrovascular events, hypoglycemia, uremia, electrolyte imbalances do not explain rapid cholinergic and GI symptoms;∘Infections—Encephalitis unlikely due to rapid onset, lack of fever, and normal inflammatory markers.

Supporting diagnostic findings include decreased cholinesterase activity and transient renal impairment, indicative of systemic toxin exposure, as well as imaging evidence of pulmonary congestion and minimal pleural effusion without infiltrates, consistent with functional-dynamic changes. Combined with acute cholinergic symptoms, CNS involvement, gastrointestinal manifestations, laboratory abnormalities, and confirmed ingestion of wild *A. muscaria*, these findings make alternative diagnoses highly unlikely. Differential considerations primarily exclude other muscarine-containing mushrooms, organophosphate poisoning, and acute metabolic or neurological disorders.

### 4.5. Mushroom Identification

No mycological examination or laboratory-based identification of the ingested mushrooms was performed, precluding definitive confirmation of the toxin. The diagnosis was established based on patient history and the observed clinical manifestations.

### 4.6. Treatment

Immediately upon admission, gastrointestinal decontamination was performed, including nasogastric intubation and gastric lavage, followed by the administration of activated charcoal (carbo medicinalis, 1 g/kg). The patients received intensive intravenous fluid therapy with electrolyte solutions (approximately 3000 mL), along with famotidine, vitamins B1 and B6, citicoline, mannitol, metoclopramide, magnesium, and subsequently forced diuresis (5000 mL/24 h) with furosemide.

Upon occurrence of a generalized seizure, emergency treatment with clonazepam (one ampule of 1 mg/mL i.v.) was administered. To control cholinergic symptoms, atropine therapy was initiated—initially as a bolus, followed by continuous infusion (0.3 mL/h, equivalent to 0.036 mg/h; 6 ampoules in a 50 mL syringe via perfusor). Additionally, dexamethasone (two ampules of 4 mg/mL) was administered. Active hydration and urine output monitoring were maintained, along with symptomatic treatment for gastrointestinal manifestations and support of vital functions.

Due to the high risk of complications—including convulsions, respiratory failure, and cardiovascular instability—strict monitoring and comprehensive therapy were implemented. The patients were transferred to the Department of Anesthesiology, Resuscitation, and Intensive Care due to the severity of their condition and the need for intensive monitoring. Atropine infusion was discontinued 18 h after hospitalization due to evident signs of atropinization.

### 4.7. Resolution of the Toxic State

Approximately 12 h after admission, both patients regained consciousness. The female patient exhibited unusual behavior, including elevated mood, vocalizations (singing), and hallucinatory experiences. Over the subsequent hours, the clinical picture stabilized, with a persistent trend toward gradual recovery. At this stage, no long-term complications were observed. Patients were advised to seek follow-up psychological support if needed, considering the potentially traumatic nature of the experience. It is not known whether they utilized this service, likely due to the reversible, short-lived, and low-disability nature of this type of mushroom poisoning.

## 5. Discussion

Mushrooms are valued for their nutritional and therapeutic properties. Of roughly 140,000 known species, over 2000 are safe to eat, and about 700 have documented pharmacological activity, though some remain toxic [38,39]. Avoiding wild mushrooms is the most effective prevention. However, in mycophilic cultures—such as Southeast Asia, the Venezuelan Amazon, Slavic countries, and Italy—mushroom foraging is culturally entrenched, unlike in mycophobic societies (e.g., the UK). Limited visual criteria for distinguishing edible from toxic species, known mainly to specialists, increase poisoning risk, leading to numerous annual intoxications [38]. Karami Matin et al. (2022) further emphasize that toxic mushrooms are among the leading etiological factors for foodborne diseases, necessitating heightened attention in public health planning [40].

Consumption of *A. muscaria* rarely leads to fatal outcomes [9]. Intoxications are largely seasonal, peaking in July–August and November, coinciding with optimal mushroom growth [18,31,41]. Seasonal variations significantly affect toxicity, with spring and summer specimens containing up to ten times more toxins than autumnal mushrooms [11]. Misidentification of species remains the primary cause of poisoning and insufficient awareness continues to be a critical risk factor despite the mushroom’s characteristic appearance and known toxicity [11,17]. Intoxications often result from intentional ingestion for suicidal or psychoactive purposes, with *A. muscaria* ranking second only to psilocybin mushrooms in such use [6,25]. A survey of 250 participants from social media communities promoting Amanita muscaria consumption revealed that primary motivations included stress reduction (29.6%), relief of insomnia (22.4%), alleviation of depressive symptoms (20.4%), pain relief (18.4%), skin improvement (5.6%), and gastrointestinal symptom relief (3.6%). Reported adverse effects were primarily lethargy, abdominal pain, and nausea. The most commonly reported forms of consumption were tinctures (24.2%), ambrosia preparations (12.1%), and dried mushrooms (6.6%) [16]. Given that processing affects bioactive compound content, further study is needed to determine its impact on clinical outcomes. Notably, in the present cases, the initial cooking water—containing water-soluble toxins—was not discarded, a precaution practiced in some countries.

Due to the rarity of mushroom poisonings and limited attention to the topic during medical training, there is a risk of delayed diagnosis and treatment. The diagnostic approach is based on three main pillars:Identification of the consumed mushroom;Interval between ingestion and symptom onset;Laboratory confirmation (where possible).

The most reliable method remains macroscopic or microscopic identification of the mushroom or its remnants by a qualified mycologist. Additional information, such as preparation method, condition, collection site, and latency to symptom onset, aids in refining the diagnosis [17]. A key limitation of the present cases is the absence of mycological or laboratory confirmation, preventing definitive identification of the toxin before treatment. While toxico-chemical analyses for qualitative and quantitative determination of mycotoxins would allow for more precise assessment, improve case management, and support antidote development, such approaches are rarely implemented due to limited data on mycotoxin metabolism and restricted access to specialized analytical instruments, primarily chromatographic [7,12,18,24,25,42]. Given substantial differences in xenobiotic profiles among species and the challenges of accurate identification, classification based on clinical manifestations is a more practical approach than strict taxonomy. This clinically oriented model facilitates diagnosis and therapeutic decision-making, reflecting the complex biochemical nature of mushroom intoxications [4].

The clinical picture observed in the two patients was characterized by severe coma, indicating profound central nervous system depression due to mycotoxin exposure. Persistent coma, accompanied by hypersalivation, miosis, and fibrillar limb fasciculations, has been documented in *A. muscaria* intoxications [11,31]. Hypersalivation reflects predominant parasympathetic activation, and together with vomiting and copious bronchial secretions, forms part of the classic cholinergic triad, also described in Inocybe and Clitocybe poisonings. While ibotenic acid and muscimol typically induce mydriasis, rare occurrences of miosis—as observed in both patients—may result from relatively higher muscarine content in the ingested mushrooms, which varies according to ecological and geographic factors [31]. Furthermore, muscimol and ibotenic acid, through agonistic action on GABAergic and glutamatergic receptors, may indirectly modulate parasympathetic activity in the brainstem, inducing central miosis under specific neurovegetative imbalances [31].

Another diagnostic consideration was whether the intoxication could involve *A. pantherina*. Despite similarities in toxicity between *A. muscaria* and *A. pantherina*, their clinical spectra differ, leading some experts to classify them as distinct toxidromes. Vendramin and Brvar (2014) noted that *A. muscaria* intoxications typically present with more pronounced confusion and agitation, whereas *A. pantherina* exhibits higher rates of coma due to elevated muscimol content [43]. Based on patient-reported details, including descriptions of the mushrooms’ external features, poisoning by *A. pantherina*—which has a less distinctive gray or brown cap with white warts—was considered unlikely. It is more probable that the toxic species was misidentified as the edible *A. caesarea*, a toxicological issue recognized since the era of Emperor Claudius. The risk of misidentification increases following heavy rainfall, as the white warts on the fly agaric cap may be washed away [31,44,45,46]. Another edible species in Bulgaria, *A. rubescens*, could theoretically be confused with *A. muscaria* [17]. Even in the absence of reliable patient-reported information, additional clinical cases, ideally supported by toxicochemical analysis, are necessary to reinforce Vendramin and Brvar’s (2014) conclusions [43]. It is well established that mushroom intoxications often produce mixed and individually variable clinical pictures, necessitating primarily supportive care with targeted therapy for muscarinic or severe central symptoms. Variations in ibotenic acid and muscimol content explain differences in onset and progression of toxicity among species [3]. Dose, route of consumption, and pre-treatment methods—such as boiling or soaking—can modulate effects, as the toxins are water-soluble yet heat-stable. Fresh mushrooms rich in ibotenic acid exhibit different toxicity compared to dried specimens, where part of the ibotenic acid is converted to muscimol. Geographic origin and season also influence neurotoxin concentrations [3,25]. Additionally, reverse conversion from muscimol to ibotenic acid is possible, catalyzed by glutamate decarboxylase, creating a dynamic equilibrium between the two metabolites, likely explaining observed interindividual variability in neurological manifestations [31].

Recent mycological research shows that traditional classifications of mushroom poisonings often overlook newly described syndromes, complicating diagnosis. Simultaneous ingestion of multiple species can produce mixed intoxication patterns. In this context, White et al. (2019) proposed a classification designed to guide diagnosis and support clinical decision-making (Table 3) [2].

We particularly value the authors’ proposed diagnostic algorithm, which aids clinicians in rapidly achieving accurate diagnoses and selecting appropriate therapies. Although assessed retrospectively, applying the algorithm among colleagues confirmed its effectiveness. Figure 5 illustrates potential clinical pathways guiding a clinician toward diagnosing Group 2C mycotoxicosis—CNS toxicity from ibotenic acid/muscimol.

The algorithm initially differentiates mycotoxicoses with short latency (<6 h); however, in our cases, hypersalivation risked misclassification as Group 2B instead of 2C. Gastrointestinal symptoms may arise from multiple species, microbial contamination, undercooked or raw mushrooms, or excessive consumption [17,47]. Determining whether mystical experiences result solely from Type 2A (psilocybin) intoxication is challenging, as *A. muscaria* is also used entheogenically. High mannitol content may enhance the brain transport of active compounds, explaining hallucinogenic effects that exceed expected doses. Co-ingestion with other psychotropic substances further influences psychoactivity [48]. In our cases, neuropsychiatric manifestations in the female patient (elevated mood, vocalizations, hallucinations) may reflect residual cholinergic effects, neurotransmitter fluctuations, or atropinization, where central anticholinergic effects—delirium, euphoria, hallucinations—are well documented [27,49,50,51]. Terminating atropine perfusion six hours after regaining consciousness was critical to prevent deepening of toxic anticholinergic syndrome and stabilize the patient. The symptom course—from rapid-onset coma and convulsions to full recovery within 24 h—aligns with typical muscarinic mushroom intoxications and supports the chosen therapy. Notably, Brvar et al. (2006) reported a case of *A. muscaria* intoxication with psychosis lasting five days, the longest documented [27].

The minimal laboratory abnormalities observed were mainly secondary to metabolic disturbances from vomiting, dehydration, hypoxia, and circulatory changes. Both patients showed no hepatotoxic or nephrotoxic effects, effectively excluding *A. phalloides* poisoning; functional changes were primarily reversible and related to metabolic stress. Clinical severity was driven by the neurotropic effects of muscimol and ibotenic acid rather than organ damage, with prognosis favorable following intensive supportive care, as reflected in laboratory recovery at discharge [7,11]. Rarely, mild reductions in cholinesterase activity may occur, although this is not considered exceptional [20].

Although gastric lavage was performed, earlier intervention (within 1–2 h post-ingestion) would likely reduce symptom severity [31]. In these cases, lavage was precautionary due to borderline timing, while symptom intensity was likely influenced by massive ingestion and delayed decontamination. CNS excitatory symptoms were effectively managed with benzodiazepines, consistent with prior reports [25,47]. Meisel et al. (2022) noted that *A. muscaria*-induced CNS toxicity often progresses from excitation to depression, highlighting the link between benzodiazepine use and potential intubation [25]. In our cases, intubation was not required, though airway readiness remains essential. The role of flumazenil in comatose patients remains unclear [17].

Prevention of mushroom intoxication requires a combined approach of education, safe foraging, and proper preparation. Measures include carrying a mobile phone or foraging with another person to ensure timely first aid and medical access. A retrospective study from the Department of Medical Toxicology in Krakow, analyzing 457 adult cases over eight years, found that 87% involved edible species, with intoxications linked to prolonged storage before or after cooking or delayed consumption, resulting in mild gastrointestinal symptoms [42]. These findings emphasize that even edible mushrooms can cause intoxication if mishandled. Key recommendations to reduce mycotoxicosis risk include:∘Consume only well-identified mushroom species; if in doubt, avoid ingestion;∘Avoid Internet-based mushroom identification by non-professionals, as morphological variability is influenced by season, growth stage, and environment;∘Store mushrooms in appropriate containers;∘Consume mushrooms immediately after cooking;∘Avoid combining mushrooms with alcohol or other psychotropic substances, particularly in recreational use, due to increased risk of central and psychiatric symptoms;∘Educate the population on signs of mushroom intoxication and the availability of regional poison centers for rapid consultation.

In addition, Table 4 summarizes the principal morphological, ecological, and molecular criteria essential for the precise differentiation of wild mushrooms. Identification relies on macroscopic traits—such as cap structure, gill morphology, and stipe characteristics—combined with microscopic examination of spores, and when necessary, DNA-based barcoding techniques. Ecological factors, including substrate specificity and mycorrhizal associations, further enhance diagnostic accuracy and facilitate the distinction of cryptic or morphologically similar taxa [52,53,54,55,56,57,58]. Implementation of these measures would significantly reduce the incidence and severity of mushroom intoxications while promoting safe foraging and consumption practices [26,42].

## 6. Conclusions

*A. muscaria* intoxication represents a distinctive and multifaceted toxicological entity, shaped by complex interactions between its psychoactive constituents, environmental variability, and evolving patterns of recreational or inadvertent use. The review highlights the central roles of ibotenic acid and muscimol in producing the characteristic excitatory–depressive neurobehavioral sequence while also clarifying the limited yet clinically relevant contribution of muscarine to the overall toxidrome. Despite generally favorable outcomes, the unpredictable severity of presentation and the potential for deep coma, seizures, and autonomic instability necessitate early recognition and rigorous supportive management.

The two clinical cases presented here—marked by rapid onset, profound central nervous system depression, cholinergic manifestations, and transient organ dysfunction—serve to reinforce and exemplify the broader toxicodynamic principles reviewed in this work. Their favorable clinical evolution underlines the effectiveness of timely symptomatic treatment, meticulous monitoring, and judicious use of atropine when muscarinic features are evident.

Taken together, the synthesized evidence and clinical observations call for enhanced toxicological awareness among healthcare professionals, improved public education on wild mushroom foraging, and further research into the metabolism, clinical biomarkers, and therapeutic strategies relevant to *A. muscaria* and related psychoactive fungi.

## Figures and Tables

**Figure 1 toxins-17-00570-f001:**
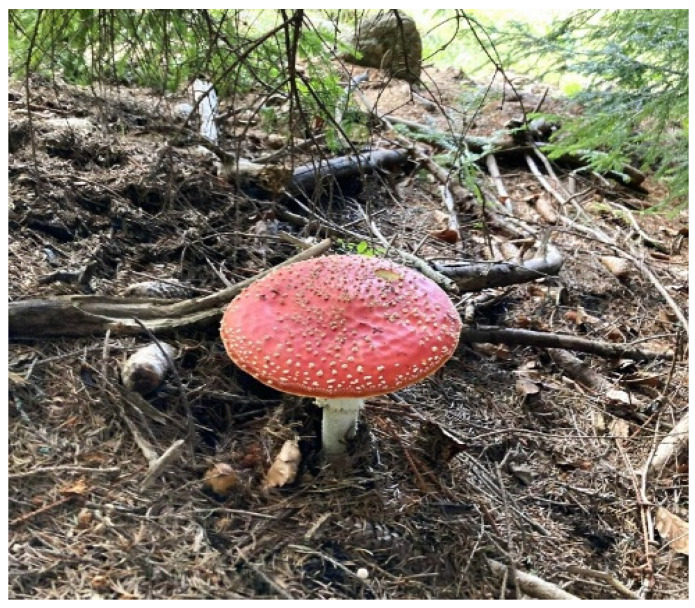
*A. muscaria* (fly agaric) in situ within a coniferous woodland, Varna Region, northeastern Bulgaria.

**Figure 2 toxins-17-00570-f002:**
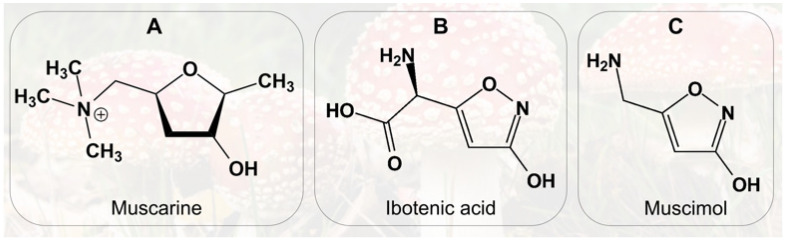
Chemical structures of: (**A**) Muscarine; (**B**) Ibotenic acid; (**C**) Muscimol.

**Figure 3 toxins-17-00570-f003:**
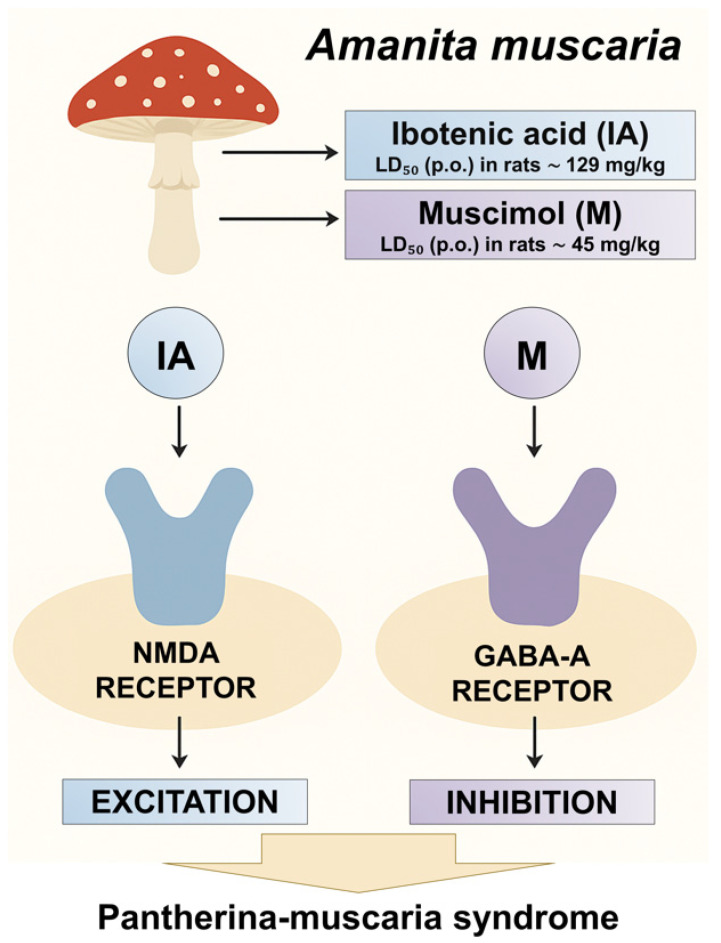
Mechanism of *A. muscaria* toxicity.

**Figure 4 toxins-17-00570-f004:**
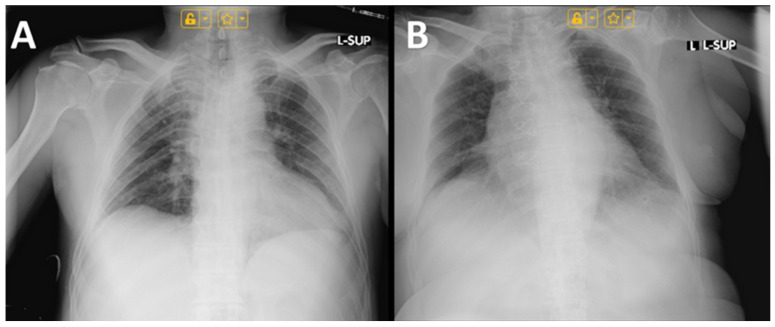
Chest radiographs of: (**A**) Male; (**B**) Female.

**Figure 5 toxins-17-00570-f005:**
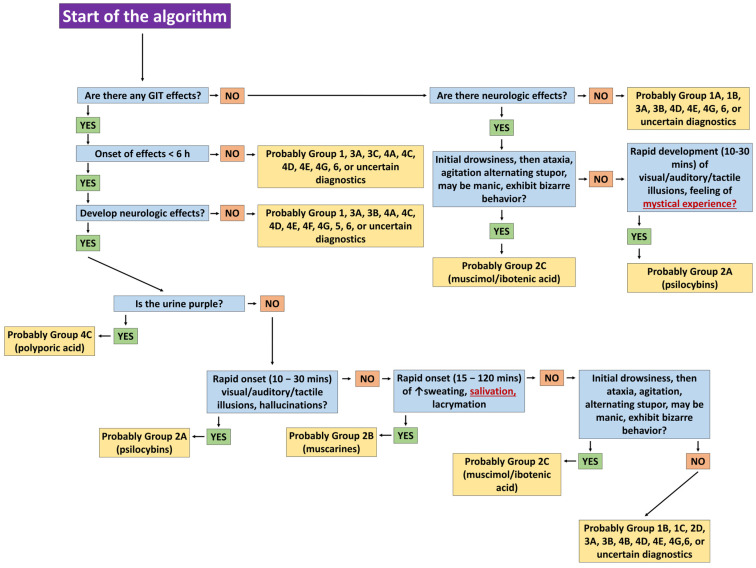
Steps in the algorithm of White et al. (2019) [2] leading to the diagnosis of mycotoxicosis from Group 2C—CNS toxicity mushrooms (ibotenic acid/muscimol).

**Table 1 toxins-17-00570-t001:** Clinical manifestations and therapeutic interventions in *A. muscaria* intoxication.

Stage/Manifestation	Clinical Features	Recommended Treatment/Interventions
Excited Phase (1–4 h post-ingestion)	∘ Sensation of warmth, tingling, and lightness ∘ Urge to move (movements rapidly become uncoordinated) ∘ Limbs perceived as “floating” ∘ Dizziness ∘ Muscle weakness ∘ Visual and auditory hallucinations ∘ Mental agitation	∘ Supportive care ∘ Monitoring of mental status ∘ Sedation if necessary
Comatose Phase (Several Hours)	∘ Deep comatose sleep with spontaneous awakening accompanied by increased neuromuscular excitability ∘ Hypotension ∘ Sensation of “reincarnation” upon awakening ∘ Headache, weakness, depressive symptoms ∘ Residual impairments in coordination, speech, and vision	∘ Sedation for agitated patients ∘ Monitoring of respiratory and cardiovascular function ∘ Supportive care
General Treatment Measures	∘ Symptomatic treatment ∘ Prognosis is generally favorable	∘ Rapid removal of ingested mushrooms from the gastrointestinal tract (vomiting, gastric lavage, activated charcoal) ∘ If needed—saline laxatives and adsorbents ∘ Early signs of muscarinic syndrome may warrant consideration of low-dose atropine

**Table 2 toxins-17-00570-t002:** Comparative overview of laboratory parameters in the two hospitalized patients at admission to the toxicology unit and at discharge.

Laboratory Parameter	Male			Female		
	Admission	Discharge	Reference Range	Admission	Discharge	Reference Range
AST (ASAT) U/L	30	19	<40	23	34	<32
ALT (ALAT) U/L	21	17	<45	11	14	<34
GGT U/L	17	17	<50	8	9	<32
Total Bilirubin µmol/L	31	24	<21	11	9	<21
Cholinesterase IU/mL	8.8	7.1	3.83–10.8	7.5	5.3	3.83–10.8
Urea mmol/L	8.6	8.2	2.8–7.2	15.5	10.4	2.8–7.2
Serum Creatinine µmol/L	95	88	62–115	132	117.0	53–106
eGFR mL/min	70	77	>55	36	41	>55

**Table 3 toxins-17-00570-t003:** Classification scheme for clinical types of mushroom poisoning.

Group	Suspected Toxin	Indicative Species
Group 1—Cytotoxic mushroom poisoning		
Subgroup 1.1—Primary hepatotoxic mushroom poisoning		
Group 1A—Primary hepatotoxicity	amatoxins	*Amanita phalloides*
Subgroup 1.2—Primary nephrotoxic mushroom poisoning		
Group 1B—Primary nephrotoxicity	AHDA	*Amanita smithiana*
Group 1C—Delayed primary nephrotoxicity	orellanine	*Cortinarius* spp.
Group 2—Neurotoxic mushroom poisoning		
Group 2A—Hallucinogenic mushrooms	psilocybins	*Psilocybe* spp.
Group 2B—Autonomic toxicity mushrooms	muscarines	*Inocybe* spp.
Group 2C—CNS toxicity mushrooms	ibotenic acid/muscimol	*Amanita muscaria*
Group 2D—Morel neurologic syndrome	unknown	*Morchella* spp.
Group 3—Myotoxic mushroom poisoning		
Group 3A—Rapid onset myotoxicity	carboxylic acid	*Russula subnigrans*
Group 3B—Delayed onset myotoxicity	saponaceolide B	*Tricholoma equestre*
Group 4—Metabolic/endocrine toxicity mushroom poisoning		
Group 4A—GABA-blocking mushroom poisoning	gyromitrins	*Gyromitra* spp.
Group 4B—Disulfiram-like mushroom poisoning	coprines	*Coprinus* spp.
Group 4C—Polyporic mushroom poisoning	polyporic acid	*Hapalopilus rutilans*
Group 4D—Trichothecene mushroom poisoning	trichothecenes	*Podostroma cornu-damae*
Group 4E—Hypoglycaemic mushroom poisoning	unusual amino acids	*Trogia venenata*
Group 4F—Hyperprocalcitoninemia mushroom poisoning	unknown	*Boletus satanas*
Group 4G—Pancytopenia mushroom poisoning	unknown	*Ganoderma neojaponicum*
Group 5—Gastrointestinal irritant mushroom poisoning		
Group 6—Miscellaneous adverse reactions to mushrooms		
Group 6A—Shiitake mushroom dermatitis	lentinan	*Lentinula edodes*
Group 6B—Erythromelalgia-like mushroom poisoning	acromelic acid	*Clitocybe acromelagia*
Group 6C—Paxillus syndrome	unknown	*Paxillus involutus*
Group 6D—Encephalopathy syndrome	Hydrocyanic acid	*Pleurocybella porrigens*

**Table 4 toxins-17-00570-t004:** Core Parameters for Accurate Differentiation of Wild Mushrooms.

Feature Category	Key Characteristics for Differentiation
Cap (pileus)	∘ Shape (convex, conical, flat) ∘ Color ∘ Surface texture (smooth, scaly, fibrillose) ∘ Margin details (inrolled, straight, uplifted)
Gills (lamellae) and spore-bearing surface	∘ Gill attachment (free, adnate, decurrent) ∘ Spacing (crowded, distant) ∘ Color changes ∘ Presence/lack of pores or tubes instead of gills
Stipe (stem) and ring/volva structures	∘ Stem length/width ∘ Base bulbous or not ∘ Presence of annulus (ring) or volva (cup at base) ∘ Texture ∘ Color contrast from cap
Spore print color and microscopic features	∘ Spore print hue (white, brown, greenish, etc.) ∘ Spore size/shape ∘ Microscopic cystidia, clamp-connections
Ecology and substrate	∘ Habitat (wood-decay, soil, mycorrhizal tree associations); substrate type; seasonality
Molecular (DNA barcode) methods	∘ Internal Transcribed Spacer region sequencing, phylogenetic placement, sequence divergence

## Data Availability

The original contributions presented in this study are included in the article. Further inquiries can be directed to the corresponding author.

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
