# Peer review of "Acute Amanita muscaria Toxicity: A Literature Review and Two Case Reports in Elderly Spouses Following Home Preparation"

_toxins, 2025, doi:10.3390/toxins17120570_

Round 1
Reviewer 1 Report
Comments and Suggestions for Authors
A good, well-written, and detailed paper making a valuabe contribution to the literature; among the more detailed case studies I have read. It would have been nice to know the intent of the victims on consuming the mushrooms (the authors speculate on the possibility of confusion with edible Amanita caesarea), but I understand that this data is likely not available. Line 164 states "Minimal quantities of amatoxins, phallotoxins, and other peptide toxins have also been reported..."; while true, these reports (not cited; I imagine Faulstich, and Butler?) have not been reproducible; and Faulstich's claims of widespread, low-concentration amatoxin distribution in a variety of mushrooms (including edible species) was refuted in a footnote in Wieland's 1986 monograph on amatoxins and phallotoxins.
Author Response
Comment 1:
A good, well-written, and detailed paper making a valuabe contribution to the literature; among the more detailed case studies I have read. It would have been nice to know the intent of the victims on consuming the mushrooms (the authors speculate on the possibility of confusion with edible Amanita caesarea), but I understand that this data is likely not available. Line 164 states "Minimal quantities of amatoxins, phallotoxins, and other peptide toxins have also been reported..."; while true, these reports (not cited; I imagine Faulstich, and Butler?) have not been reproducible; and Faulstich's claims of widespread, low-concentration amatoxin distribution in a variety of mushrooms (including edible species) was refuted in a footnote in Wieland's 1986 monograph on amatoxins and phallotoxins.
Response 1:
The intent of the affected individuals in consuming the mushrooms was presented in a speculative manner, due to the accidental nature of the poisoning and the limited data arising from the victims’ insufficient awareness. The misleading sentence concerning the content of amatoxins, phallotoxins, and other peptide toxins has been revised.
Reviewer 2 Report
Comments and Suggestions for Authors
This is an interesting study though the novelty is on low side. There are number of studies already available in the literature on the subject but still the manuscript provide a combination of literature review and couple of case reports. I have few suggestions to make for the improvement of manuscript -
- Authors seem confused if they are preparing a review paper of focusing on two case reports. In my opinion their objective should be clear. Abstract does not tell anything about literature review.
- Section 1, 2 and 3 reads more like a review paper summarising the literature on the subject (nearly 6 pages single space) while the two case reports starts only after page 7. In my opinion authors have dragged their manuscript unnecessarily and need to be more focussed.
- Authors need to suggest how to differentiate between various forms of mushroom in a more precise manner.
- Discussion again to too long and mostly talked about something which is known for years. It lacks focus and must be shortened considerably.
- Same applies to the conclusion part which again seems to be focused on two case report. Conclusion need to be shortened, focused and must be listed pointwise.
Author Response
Comment 1:
Authors seem confused if they are preparing a review paper of focusing on two case reports. In my opinion their objective should be clear. Abstract does not tell anything about literature review.
Response 1:
Information has been added to both the abstract and the manuscript indicating that, in addition to the clinical cases, the study also provides a literature review.
Comment 2:
Section 1, 2 and 3 reads more like a review paper summarising the literature on the subject (nearly 6 pages single space) while the two case reports starts only after page 7. In my opinion authors have dragged their manuscript unnecessarily and need to be more focussed.
Response 2:
The text in Sections 1, 2, and 3 was condensed in a manner that preserves the integrity and completeness of the manuscript’s overall content.
Comment 3:
Authors need to suggest how to differentiate between various forms of mushroom in a more precise manner.
Response 3:
Core parameters for accurate differentiation of wild mushrooms has been added into the text.
Comment 4:
Discussion again to too long and mostly talked about something which is known for years. It lacks focus and must be shortened considerably.
Response 4:
The Discussion section has also been shortened.
Comment 5: Same applies to the conclusion part which again seems to be focused on two case report. Conclusion need to be shortened, focused and must be listed pointwise.
Authors seem confused if they are preparing a review paper of focusing on two case reports. In my opinion their objective should be clear. Abstract does not tell anything about literature review.
Response 5:
The Conclusion section was shortened and more focused on the reported clinical cases.
Reviewer 3 Report
Comments and Suggestions for Authors
In this manuscript, authors have studied about two contemporaneous cases of acute A. muscaria intoxication in elderly spouses following ingestion of home-prepared wild mushrooms and reported that after ingestion both patients developed rapid-onset gastrointestinal disturbances, pronounced neurological deficits, hallucinations, and transient comatose states. It is a good piece of research for human welfare. The submitted manuscript in its current form is not acceptable for publication in the esteemed “Toxins” journal and need minor revision. Please refer to my following suggestions to further improve this manuscript.
Other Comments:
- Please add reference range of laboratory parameters in Table 2.
- Mushroom scientific name should be in italic throughout manuscript.
- Introduction is looking lengthy. It is suggested that introduction section can be reduced.
- Authors can add Figure/Figures for diagrammatic representation of mechanism of toxicity.
- A paper has been published on the almost same objectives. Please justify the submitted paper in terms of novelty.
Meisel EM, Morgan B, Schwartz M, Kazzi Z, Cetin H, Sahin A. Two Cases of Severe Amanita Muscaria Poisoning Including a Fatality. Wilderness & Environmental Medicine. 2022;33(4):412-416. doi:10.1016/j.wem.2022.06.002
- Ethical approval is missing. Please mention the approval number by the ethical committee.
- Are any ultrasound or MRI reports of patients available to find some anomalies in the organs structure? If available, authors can add. It could be significant findings in addition to reported data.
Author Response
Comment 1: Please add reference range of laboratory parameters in Table 2.
Response 1: Reference range of laboratory parameters have been added in Table 2.
Comment 2: Mushroom scientific name should be in italic throughout manuscript.
Response 2: Mushroom scientific name has been made in italic throughout manuscript.
Comment 3: Introduction is looking lengthy. It is suggested that introduction section can be reduced.
Response 3:
The introduction section has been reduced.
Comment 4: Authors can add Figure/Figures for diagrammatic representation of mechanism of toxicity.
Response 4:
A figure illustrating the mechanism of A. muscaria toxicity has been generated (Figure 3).
Comment 5: A paper has been published on the almost same objectives. Please justify the submitted paper in terms of novelty. Meisel EM, Morgan B, Schwartz M, Kazzi Z, Cetin H, Sahin A. Two Cases of Severe Amanita Muscaria Poisoning Including a Fatality. Wilderness & Environmental Medicine. 2022;33(4):412-416. doi:10.1016/j.wem.2022.06.002
Response 5:
In Meisel et al. (2022), two independent clinical cases are reported, one of which was fatal. As described in the manuscript, our cases of a 78-year-old male and a 79-year-old female, provide a unique opportunity to analyze the toxicokinetic and toxicodynamic characteristics of A. muscaria intoxication under identical exposure, while highlighting individual differences in physiological response. Simultaneous symptom onset supports a clear causal link and illustrates age-related susceptibility. Clinical data confirm predominant neurotoxic effects with minimal organ involvement, emphasizing the importance of careful assessment, prompt supportive care, and public awareness.
Comment 6: Ethical approval is missing. Please mention the approval number by the ethical committee.
Response 6:
We would like to clarify that this study is retrospective and relies solely on medical data that had already been collected for other clinical purposes. Therefore, no approval was required from an ethics committee or an institutional review board, as no new interventions were performed on patients and all data were obtained from existing medical records.
Comment 7: Are any ultrasound or MRI reports of patients available to find some anomalies in the organs structure? If available, authors can add. It could be significant findings in addition to reported data.
Response 7:
All types of conducted investigations have already been described in the manuscript.
Round 2
Reviewer 2 Report
Comments and Suggestions for Authors
Though authors have made significant efforts in modifying their manuscript but still I feel the title and the text primarily should focus on review supported by two case reports. In the present form, while most of the text focus on review, abstract and conclusion primarily discuss about the two case reports. Title may be "A review of Acute Amanita muscaria Toxicity in Elderly Spouses", Abstract and Discussion need to be on review of literature, lacunas and future direction
Author Response
Dear Editors,
Dear Reviewer 2,
We sincerely thank you for recognizing our efforts to improve the quality of our work!
Here are the responses needed (Please note that all corrections have been highlighted in yellow in the text):
Comment 1:
Though authors have made significant efforts in modifying their manuscript but still I feel the title and the text primarily should focus on review supported by two case reports. In the present form, while most of the text focus on review, abstract and conclusion primarily discuss about the two case reports. Title may be "A review of Acute Amanita muscaria Toxicity in Elderly Spouses", Abstract and Discussion need to be on review of literature, lacunas and future direction
Response 1:
The title and the text (including the Abstract, the Discussion, and the Conclusion) have been revised to be more focused on a review supported by two case reports.